# Bolsa Familia Program and Perinatal Outcomes: NISAMI Cohort

**DOI:** 10.3390/ijerph19095345

**Published:** 2022-04-28

**Authors:** Cinthia Soares Lisboa, Jerusa da Mota Santana, Rita de Cássia Ribeiro-Silva, Edna Maria de Araújo, Carlos Alberto Lima da Silva, Mauricio Lima Barreto, Marcos Pereira, Djanilson Barbosa dos Santos

**Affiliations:** 1Programa de Pós-Graduação em Saúde Coletiva, Feira de Santana State University, Av. Transnordestina, s/n, Feira de Santana, Novo Horizonte 44036-900, BA, Brazil; ednakam@gmail.com (E.M.d.A.); carlosls.compos@gmail.com (C.A.L.d.S.); 2Center of Health Sciences, Universidade Federal do Recôncavo da Bahia, Santo Antônio de Jesus 44574-490, BA, Brazil; jerusanutri@gmail.com; 3School of Nutrition, Federal University of Bahia, Salvador 40170-110, BA, Brazil; ritaribeiroufba@gmail.com; 4Collective Health Institute, Universidade Federal da Bahia, Rua Basílio da Gama, s/n, Salvador 40110-040, BA, Brazil; mauricio.barreto@bahia.fiocruz.br; 5Center for Data and Knowledge Integration for Health (CIDACS), Oswaldo Cruz Foundation, Salvador 41745-715, BA, Brazil

**Keywords:** pregnancy, premature birth, birth weight, child health, nutrition programs and policies, social programs

## Abstract

This study aimed to assess the impact of the Bolsa Familia Program on perinatal outcomes of pregnant women. A cohort study was conducted with pregnant women supported by prenatal services at 17 Family Health Units in Bahia, Brazil. A previously tested structured questionnaire, which has sociodemographic, economic, prenatal care, lifestyle, and nutritional variables, has been used to collect data. The outcomes included premature birth and low birth weight. A hierarchical conceptual model was constructed, and logistic regression analysis was performed. From a total of 1173 pregnant women, the identified average age was 25.44 years and 34.10% had pre-gestational overweight. The non-beneficiary pregnant women presented a 1.54 (95% CI = 0.46–5.09) times higher chance of giving birth to children with low weight and a 1.03 (95% CI = 95% CI = 0.53–2.00) times chance of premature birth when compared to the beneficiary group. In the multilevel model, some variables were statistically significant, such as age between 18 and 24 years (*p* = 0.003), age greater than or equal to 35 years (*p* = 0.025), family income (*p* = 0.008), employment status (*p* = 0.010), and maternal height (*p* = 0.009). The Bolsa Familia Program, as an integrated strategy of social inclusion and economic development, is suggested to exert a protective effect on the health of mother-concept binomial.

## 1. Introduction

Gestation causes physiologic modifications on the maternal organism, demanding a higher need for essential nutrients. Inappropriate energetic intake of pregnant women may lead to competition between mothers and fetuses, restricting the availability of necessary nutrients for adequate fetal growth. In this sense, the literature is consensual to recognize that the maternal nutritional status is an indicator of health and quality of life for women and the growth of their children, especially for birth weight, once the unique source of nutrients of the concept is constituted by maternal nutritional supply and dietary intake [1].

Low Birth Weight (LBW, birth weight lower than 2.5 kg) is considered one of the most prominent public health issues, mostly in Brazil, due to the impact on infant morbidity and mortality. The prematurity and the intrauterine growth restriction have been indicated as the major factors for this condition [2].

In Brazil, the prevalence of LBW is 9.2%, which may vary depending on the region. However, the most dramatic situations are presented by the Northern (12.2%) and Northeastern (12.0%) regions of Brazil, with the prematurity prevalence of 11.7% and 11.8%, respectively, in the three-year term 2009–2011, caused among other things by inadequate maternal dietary intake and difficulties regarding the access of health services [3].

Concerned by these indicators, the Brazilian Federal Government, through the Provisional Measure N° 132, implemented the Bolsa Família Program (BFP) in 2003, as a national program of conditional cash transfer, intended for impoverished and extremely impoverished families, having as its aim the insurance of promoting access to the public services network, particularly in health, education and social protection; combating hunger and promoting food and nutrition security (FNS); and stimulating the sustained empowerment of families living in poverty and extreme poverty [4].

The BFP seeks to invest in human capital, by associating conditional cash transfer with educational goals and uptake of health services. Among the conditionalities, the family must fulfil the health and education conditions to maintain the program: a minimum school attendance of 85.0% of the monthly school hours for children seven to seventeen years old, a health and nutrition agenda for beneficiary families with pregnant women, nursing mothers or children under seven years of age (prenatal care, vaccination, health and nutrition surveillance) [4,5].

Every two years the information in the Unified Registry (CadÚNICO) must be updated with each family’s most recent address, employment, and socioeconomic status. This activity is organized by the local social services, and family payments can be suspended or cancelled if there is evidence of child labour, fraud or the deliberate provision of incorrect information, if the family’s socioeconomic conditions improve or in the absence of compliance with the established conditions [4,5,6].

Among the proximal determinants that affect perinatal mortality (fetal deaths after the 22nd gestational week) and neonatal mortality (neonatal deaths before seven days of life) are prematurity and LBW, trigged by factors of risk, such as past pregnancy history, maternal behavior, both attendance and quality of prenatal medical assistance, socioeconomic status, and others [1,7].

Adverse socioeconomic status, such as low education level and low-income families, leads to a gestational risk triggering, once this status is generally associated with stress and worse nutritional conditions [2]. Given what has been exposed, the gestational outcomes evaluated in this study are probably the most relevant isolated factors associated with neonatal, postneonatal, and infant mortality, as well as infant morbidity and risk of many diseases in adult age. Furthermore, this is a first time, when perinatal outcomes are evaluated as connected to BFP in the Recôncavo of Bahia.

In this regard, this study aims to verify the effects of BFP on perinatal outcomes, including premature birth and low birth weight.

## 2. Materials and Methods

### 2.1. Study Design

This dynamic prospective cohort study approaches pregnant women included on the baseline of the Research Center in Maternal and Child Health (NISAMI) cohort—Maternal factors of risk to low birth weight, prematurity, and delays in intrauterine fetal growth of women enrolled in Family Health Units (USF) from Recôncavo da Bahia—in Santo Antônio de Jesus, spanning the period from 2012 to 2018.

The municipality had 21 Family Health Team (ESF) by the period of research, from which 17 ESFs were located in urban areas and included in this study. The others were considered ineligible to compose the sample for being located in neighbourhoods of difficult access or situated in the rural zone due to the insufficiency of human and economic resources.

### 2.2. Exclusion and Inclusion Criteria

The studied population comprises adult pregnant women, 18 years old or older, residing in the urban zone, presenting any gestational age, and enrolled in USFs. Excluded from sampling are pregnant teenagers, women with multiple gestations, women with Human Immunodeficiency Virus (HIV), and others who had the benefit cancelled or received it after pregnancy.

### 2.3. Data Sources

The data collection occurred in three stages: the first one consisted of selecting pregnant women from prenatal service units where a semi-structured questionnaire was applied, surveying information regarding socioeconomic status, demography, health, obstetric, and access to social welfare programs. After training from September 2011 to November 2012, students collected data for this survey.

The questionnaire was structured in seven blocks, including the following variables, namely: Information on identification and socioeconomic data: name, address, date of birth, education level, marital status, race/color, profession, issues related to family income, religion, participation in social programs; Lifestyle information: smoking, use of alcohol and other drugs; Nutritional information: pre-gestational weight and prenatal nutritional care; Gynecological and obstetric information: date of menarche; number of pregnancies and deliveries; time, type and place of last delivery; prenatal care (number of consultations, beginning of prenatal care), self-reported complications (gestational diabetes, hypertension, asthma, night blindness); use of antianemic; history of miscarriage, haemorrhage, blood transfusion. Laboratory tests: blood glucose, blood count, HIV, cytomegalovirus and parasitological stool; Medication information: use of medication in the pre-gestational and gestational period; reason for use and indication; Information on sun exposure: frequency and duration of sun exposure, use of sunscreen.

The second stage occurred at the Luís Argolo Maternity Hospital (HMLA), where the Newborn (NB) has undergone anthropometric measurements (weight, height, and circumference), which a trained nursing staff of HMLA has performed. The child was entirely undressed when weighed through a Welmy^®^ digital pediatric scale, having a capacity of 15 kg and an accuracy of 10 g, and measured in length through a Wiso^®^ compact stadiometer. The latter was measured by carefully laying the child with its head placed on the fixed component of the anthropometer and sliding the moving segment to the feet.

The weight and the length were within maximum variations of 10 g for the former and 0.1 cm for the latter, both primarily undergone a double-check. During the occurrence of variations above the limits, a third measurement was necessary. The final result was the average of the nearest measured values.

The third stage occurred in 2018 through daily visits to the Municipal Office of Social Assistance to check the Benefícios do Cidadão System—SIBEC. Initially, the CadÚNICO—SIDUN was accessed, searching for families by using the beneficiary’s name, her date of birth, and her mother’s name (required fields). The Number of Social Identification (NIS) was found through this search. Afterwards, the CadÚNICO was once more accessed, selecting the municipality of the survey. Eventually, through a search on SIBEC, selecting the option of consultation by family, and entering the found NIS, the history of the beneficiary’s situation was obtained.

### 2.4. Data Management and Statistical Analyses

This study considers the prevalence of 9.8% for LBW and 7.5% for prematurity to define the sampling size. By considering a loss of 15%, a minimum of 800 pregnant women was necessary to find correlations between the maternal risk factors and the gestational outcomes [6]. The power of the study calculated is rooted in the prevalence of beneficiary women of PBF residing in Brazil’s Northeastern region within stratification of an income lower than $12.20 [4].

The study considers the LBW and the prematurity as the outcome variables and classified them into [(0) appropriate (≥2500 g); (1) inappropriate (≤2500 g)] and [(0) ≥37 gestational weeks; (1) ≤37 gestational weeks)], respectively.

The receiving of BFP income is the central exposure variable of the study and is classified into [(0) Yes; (1) No]. Additional covariables are maternal age [18–24 years old (0); 25–34 years old (1); ≥35 years old (2)], family income [≥2 minimum salaries (0); ≤2 minimum salaries (1)], education level [≥high school (0); <high school (1)], marital status [with a partner (0); without a partner (1)], employment status [active (0); inactive (1)], possession of items [≥22 points (0); ≤22 points (1)], smoking [Yes (1); No (0)] alcohol consumption [Yes (1); No (0)], pregestational anthropometric status [low weight (0); appropriate (1); overweight (2)], maternal height [≥1.50 m (0); <1.50 m (1)], number of prenatal attendance [≥6 consultations (0); <6 consultations (1)], access to nutritional orientation [Yes (0); No (1)], and previous children with LBW [(0) No; (1) Yes].

Firstly, a descriptive analysis was performed to characterize the pregnant women profile, and a four-stage hierarchical model was adopted, according to the distal-intermediate-proximal patterns of the problems under investigation.

A bivariate analysis was conducted, and the statistically significant variables (value of *p* ≤ 0.20) were selected according to their respective levels for the multivariate analysis with Logistic Regression [8].

Thus, all the variables on every level of the hierarchy and potentially related to LBW and prematurity were included in the models, then gradually removed until reaching the significant point of *p* ≤ 0.05. Odds Ratio (OR) was applied as a measure of association. We employed the Stata software (STATA™), version 12, for the statistical analyses.

## 3. Results

### 3.1. Cohort Characteristics

The characterization of sociodemographic, economic, prenatal healthcare, lifestyle, and nutrition aspects during gestation regarding PBF are as presented in Table 1. A total of 1173 pregnant women enrolled in prenatal services of the public health system participated in this study.

Concerning the socioeconomic aspects, the sample age ranged from 18 to 24 years old, with an average of 25.44 (DP ± 6.20). A family income lower or equal to two minimum salaries is the situation for 69.52% of women. Most of them were non-beneficiary of BFP (90.65%), black-skinned (84.13%), and half of them were unemployed (52.7%).

### 3.2. Bolsa Familia Program and Perinatal Outcomes

In this study, the incidences of low weight and prematurity were 4.68% (N = 44) and 12.02% (N = 113), respectively, in which there was a positive association of the non-beneficiary women with the previously mentioned outcomes (OR = 1.54; 95% CI = 0.46–5.09), (OR = 1.03; 95% CI = 0.53–2.00). In other words, the non-beneficiary pregnant women presented a 1.54 times chance of giving birth to children with low weight and a 1.03 times chance of premature birth compared to the beneficiary group, but this association is not statistically significant.

In the bivariate analysis (Table 1), the results demonstrate that BFP presented a statistically significant association with maternal age (*p* = 0.020), family income (*p* = 0.008), employment status (*p* = 0.009) and anthropometric pregestational status (*p* = 0.040).

Appendix A and Appendix A present the logistic regression models for the LBW and prematurity outcomes, respectively, according to maternal characteristics of nutrition and lifestyle. The results demonstrate the decomposition of the estimated total effect, non-mediated (direct) and mediated determiners, obtained by adjusting three regression models, according to the conceptual model previously defined.

The variables with values of *p* ≤ 0.20 in the gross analysis participated in the hierarchical model. Therefore, the total effect of Appendix A was statistically significant for age between 18 and 24 years old (*p* = 0.003), age elder than or equal to 35 years old (*p* = 0.025), family income (*p* = 0.008), employment status (*p* = 0.010), and maternal height (*p* = 0.009). Appendix A presents the effect of associations obtained through all three models. The last model remains with the variables age between 18 and 24 years old (*p* = 0.002), age elder than or equal to 35 years old (*p* = 0.940), and skin color (*p* = 0.018).

## 4. Discussion

This research provides relevant information about the benefits of BFP on pregnant women and perinatal outcomes. The BFP, as an integrated strategy of social inclusion and economic development, implies protector effects on the health of pregnant women in the studied municipality. Thus, the results indicate that the beneficiaries have a lower probability of having premature children or low birth weight.

Along with Brazil, two other countries of Latin America (Colombia and Mexico) have presented similar results by implementing programs of income transfer, where there has been a positive impact on the health and nutrition of children younger than five years old belonging to the most impoverished families in the study [9]. Accordingly, Baber and Gertler [10] have observed that low birth weight decreased by 4.6% among the beneficiary women in Mexico.

In the light of the above, BFP may be interpreted as a program focused not only on direct income transfer to families, intending to mitigate the poverty in the short term, but also on the conditionalities that encourage the beneficiary women to attend the education and health services, therefore contributing to the improvement of health conditions, such as the increased use of preventive healthcare services, immunization coverage, and engagement with healthy practices [11,12].

Among the conditionalities of this program, those directed to the studied cycle are emphasized, as they ensure the access of pregnant women to the prenatal and postpartum consultations and the participation in actions of nutritional and dietary education from the primary health care system, especially the Family Health Program [12]. However, 10.21% of beneficiary women have undergone less than six consultations during prenatal.

According to Rasia & Albernaz, considering the advancements related to the increase of prenatal service coverage in Brazil, it is still possible to identify inequality in prenatal healthcare offered to pregnant women. Those in an adverse socio-economic situation and of low education levels are susceptible to factors associated with worse health and nutritional conditions—women who are less likely to search for prenatal assistance, evidencing the social inequality, which affects the access to healthcare services, such as prenatal, according to the inverse care law [13,14].

The influence of other social anthropologic factors of impression and expression on the non-attendance of some pregnant to the prenatal services must be considered, apart from the epidemiologic aspects. For this purpose, it is necessary to comprehend their meanings and social representations about gestation and the non-attendance to prenatal services. Thus, an integrative review revealed that the association of both encompassed sociocultural, emotional, and family aspects [15,16].

A retrospective cohort from the USA demonstrated that women with inappropriate prenatal healthcare had a higher risk of premature birth (OR = 2.0, IC 95%; 1.9–2.0), LBW (OR = 1.7; IC 95% 1.6–1.7), and infant mortality (OR = 1.5; IC 95% 1.3–1.7) compared to women who had appropriate prenatal monitoring [17].

It is necessary to emphasize that during prenatal care, nutritional education is available as a part of the assistance, despite being observed that, in this research, 51.67% of pregnant women have received no dietary orientation [18,19].

Studies suggest that when pregnant women receive nutritional orientation, their nutritional status improves for both underweight and overweight groups of women. In other words, a shift in dietary behavior is related to access to nutritional education during gestation [19,20,21].

Inadequacy in maternal nutritional status during both pregestational and gestational periods contributes to the emergence of gestational intercurrence that negatively influences the progress of gestation [22]. Some of the intercurrences, such as changes in weight gaining, are associated with a higher rate of perinatal and maternal morbimortality, as well as higher abortion risk, prematurity, and LBW [23,24].

The study conducted with 531 adult pregnant women in Brazil indicated a high prevalence of obesity (9.2%), overweight (22.7%), and low weight (25.9%) among beneficiaries of BFP [20]. These outcomes are different from those found in this research, in which 8.9% of the same group presented appropriate pregestational BMI. In parallel, an investigation conducted in the Recôncavo da Bahia region in the same population identified that inclusion in BFP implied a direct negative effect on BMI during gestation [20].

As previously mentioned, the literature considers adverse socioeconomic situations, such as low education levels and low family income, ignite gestational risks, which are associated with stress and worse nutritional conditions [25]. In this type of population, perinatal outcomes have been worse in terms of perinatal death, prematurity, congeneric anomalies, and LBW [24].

Therefore, it is necessary to stress that BFP is an integrated strategy of social inclusion and economic development, as well as the consolidation of social policies in education, health, employment, and social assistance fields, which contributed to the mitigation of poverty and hunger in Brazil [26], and consequently the infant mortality [22,27].

Thus, the existence of a regular income does not mean a guarantee against difficulties related to poverty: such problems need to be previously solved through public policies aiming to meet basic needs, whether directly (provision of primary services) or indirectly (providing conditions in which individuals may afford their basic needs) [26].

The literature affirms that beneficiary families primarily spend the BFP income on food acquisition [20]. Studies indicate that family expenses are strongly related to the perception of this income as a bonus essentially allocated to afford their children, and most women used this benefit for food, education, and clothing them [26].

The improvement in the health status of children was an expected result of BFP. Among the most relevant influences, a decrease in LBW prevalence is on focus, once it is one of the main factors associated with infant mortality, having a significant reduction even among mothers with low education levels [24].

It is well known that low levels of education are related to infant morbimortality, growth restriction, undernutrition, lower chance of attending to more than six prenatal consultations by pregnant women, greater struggling to comply with the vaccination calendar, and a higher risk of cardiovascular disease or premature mortality [28].

Race/skin color is recognized as a representative variable of inequalities in health [29]. This study found a significant statistical association between black women and premature outcomes. Cohort studies performed in the Bahia region and the USA identified that black women had a respective 51% and 21% higher risk of premature birth when compared to non-black women [30,31].

Another variable regarding socioeconomic inequalities is women’s employment status, in which 52.7% of women were inactive by the moment when interviewed, and 12.5% of them were beneficiaries of BFP. These facts diverge from the studies [30] that indicate slightly higher participation of beneficiaries in the labor market and a small reduction of hours worked of mothers. These effects, when observed, generally occurred on a small scale [32].

The studies that depict the association of BFP and perinatal outcomes with robust design methodology and approach of primary data are incipient. From this perspective, the present investigation makes progress through this gap by studying at a longitudinal level.

From the methodologic perspective, it is necessary to consider that even statistically adjusting the models of this study, investigations with observational study design present some limitations. However, this research has used a more robust strategy of analysis, which is hierarchical analysis, taking as fundament a conceptual model previously defined, which made possible a better control of the confounding factors and measurement of decomposition of total effect into their non-mediate components (or direct) in the investigated connections.

The authors point out that the BFP started in 2003 was replaced at the end of 2021 for Auxílio Brazil, through provisional measure 1061, of August 2021. There are still many uncertainties about the new income transfer program, which has consequences to the most vulnerable groups. The BFP was a real factor in the lives of pregnant women during data collection.

## 5. Conclusions

Based on the data in the analysis, this study found evidence of the positive impact that PBF fulfill in this life cycle, indicating as protector factor to the health of the binomial mother-concept, among other benefits, better nutritional assistance, contributing to the activities of the advancement of health and prevention to risk factors.

The identification of the associated local characteristics may help healthcare teams to plan more effective interventions on groups of women under more vulnerabilities, mitigating, consequently, infant morbimortality.

## Figures and Tables

**Table 1 ijerph-19-05345-t001:** Distribution of sociodemographic, economic, prenatal, lifestyle and nutritional characteristics according to the social protection program during pregnancy. Santo Antônio de Jesus, Bahia. 2011–2018.

Variables	N	%	Bolsa Família Program				*p*-Value *
			No		Yes		
			N	%	N	%	
**Distal Level**							
**Maternal Age (*n* = 1173)**							0.020
18–24 years old	552	47.06	514	93.12	38	6.88	
25–34 years old	528	45.01	468	88.64	60	11.36	
≥35 years old	93	7.93	81	87.10	12	12.90	
**Income (*n* = 899)**							0.008
≥2 MS	274	30.48	258	94.16	16	5.84	
≤2 MS	625	69.52	553	88.48	72	11.52	
**Skin Color (*n* = 939)**							0.535
Non-black women	149	15.87	132	88.59	17	11.41	
Black women	790	84.13	713	90.25	77	9.75	
**Education Level (*n* = 928)**							0.109
≥High school	211	22.74	196	92.89	15	7.11	
<High school	717	77.26	639	89.12	78	10.88	
**Marital Status (n = 935)**							0.911
With a partner	778	83.21	701	90.10	77	9.90	
Without a partner	157	16.79	141	89.81	16	10.19	
**Employment Status (*n* = 926)**							0.009
Active	438	47.30	406	92.69	32	7.31	
Incative	488	52.70	427	87.50	61	12.50	
**Intermediate Level I**							
**Smoking (*n* = 928)**							0.361
No	595	64.12	540	90.76	55	9.24	
Yes	333	35.88	296	88.89	37	11.11	
**Alcohol Comsuption (*n* = 935)**							0.702
No	257	27.49	233	90.66	24	9.34	
Yes	678	72.51	609	89.82	69	10.18	
**Intermediate Level II**							
**Possession of Items (*n* = 642)**							0.370
>22 points	99	15.42	92	92.93	7	7.07	
≤22 points	543	84.58	489	90.06	54	9.94	
**Proximal Level**							
**Pregestational Anthropometric Status (*n* = 830)**							0.040
Low weight	64	7.71	62	96.88	2	3.13	
Appropriate	483	58.19	440	91.10	43	8.90	
Overweight	283	34.10	247	87.28	36	12.72	
**Maternal Height (*n* = 854)**							0.353
≥1.50 m	803	94.03	725	90.29	78	9.71	
<1.50 m	51	5.97	44	86.27	7	13.73	
**Number of Prenatal Consultations (*n* = 888)**							0.074
≥6	46	5.18	45	97.83	1	2.17	
<6	842	94.82	756	89.79	86	10.21	
**Nutritional Orientaion (*n* = 898)**							0.186
Yes	434	48.33	396	91.24	38	8.76	
No	464	51.67	411	88.58	53	11.42	
**Children with LBW (*n* = 407)**							0.579
No	346	85.01	288	83.24	58	16.76	
Yes	61	14.99	49	80.33	12	19.67	

* ≤0.05.

## Data Availability

Not applicable.

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
