# Peer review of "Bolsa Familia Program and Perinatal Outcomes: NISAMI Cohort"

_ijerph, 2022, doi:10.3390/ijerph19095345_

Round 1

Reviewer 1 Report

The study is centred on a relevant topic and the authors have an interesting approach. However, I believe the manuscript could be improved to incorporate more suitable information for an international reader. I suggest the following points to assist the author(s):

Introduction

'Bolsa Família' is a relevant programme in the social policies scope, that has a focus on social protection for families in extreme poverty, integrating benefits in areas such as social assistance, health, among others.  However, the legal and theoretical framework regarding this programme is missing in this study. I suggest that this section be further developed. You may also create a new section that introduces 'Bolsa Família' (especially in health outcomes, since that is your main topic). There are several studies and other governmental information sources that will help with this point. In my opinion, completing this section is essential to explain the context of this study.

Material and Methods

You could provide more information about the questionnaire structure and data collection procedures. I also consider it important to explain the link between the three stages of this study since they are different and intervalled.

Conclusion

Your study has interesting findings but could benefit with more developments in the conclusions, making its implication stronger in this section. 

It is a nice and well-done work but it is worth re-working your manuscript to show the full value of this research.

Author Response

Comments and Suggestions for Authors

The study is centred on a relevant topic and the authors have an interesting approach. However, I believe the manuscript could be improved to incorporate more suitable information for an international reader. I suggest the following points to assist the author(s):

Introduction

'Bolsa Família' is a relevant programme in the social policies scope, that has a focus on social protection for families in extreme poverty, integrating benefits in areas such as social assistance, health, among others.  However, the legal and theoretical framework regarding this programme is missing in this study. I suggest that this section be further developed. You may also create a new section that introduces 'Bolsa Família' (especially in health outcomes, since that is your main topic). There are several studies and other governmental information sources that will help with this point. In my opinion, completing this section is essential to explain the context of this study.

Answer: We appreciate the comments and careful reading of our article. We performed the requested revision

Material and Methods

You could provide more information about the questionnaire structure and data collection procedures. I also consider it important to explain the link between the three stages of this study since they are different and intervalled.

Answer: we insert more information about the questionnaire structure and data collection procedures

Conclusion

Your study has interesting findings but could benefit with more developments in the conclusions, making its implication stronger in this section. It is a nice and well-done work but it is worth re-working your manuscript to show the full value of this research.

Answer: we improved the conclusions.

Reviewer 2 Report

The topic of this article is among the most important ones in human and social perspectives as it deals with the health of pregnant women and newborn babies. The scene is in Brazil where the numbers of infant morbidity and mortality are high. The research reported in the manuscript evaluates the impact of a family program (Bolsa Familia Program BFP) on pregnant women of a named cohort.

The research aim is defined in Introduction as verifying the effects of BFP on perinatal outcomes, including premature birth and low birth weight.

It is easy for the reader to understand the basic idea of the article and the main results reported.

The research aims as such are clear and from medical point of view relevant literature on perinatal outcomes is used to justify the chosen research approach.

However, other evaluations of Bolsa are not brought forward when justifying this individual research. The reader does not know whether this is a first time, when perinatal outcomes are evaluated as connected to Bolsa.

In addition, there are some details that need revision in general, but especially from the point of view of international readership.

The title of the article “Bolsa Familia Program and perinatal outcomes: NISAMI cohort” includes two phenomena that should be defined and explained clearly and thoroughly.

Bolsa Familia Program (BFP) is mentioned shortly in Introduction, but the reader is not provided with all necessary information about it and the services it offers to citizens. Some features of BFP are mentioned only in Discussion section, together with references about other services of which the reader does not know, whether they belong to Bolsa or to other systems. The content of conditionality of Bolsa is not explained at all. Usually, the conditionality in welfare programs like this is supposed to be among the most significant mechanisms that secure the desired impact. Also, Discussion section includes some international references that could be used when presenting the research aim in its social surroundings.

Also, the reader needs to know whether this  program, started in 2003, is still going on. There have been changes in the Brazilian government, raising a question about Bolsa’s survival. In this case Bolsa was a real factor in the lives of the pregnant mothers during data gathering and this cannot be changed. What about Discussion section? It should be up to date.

NISAMI is explained in 2. Materials and Methods, 2.1. Study Design. However, it is difficult to understand, what NISAMI actually is. The same can be said about other abbreviations like SIBEC and CadUNICO-SIDUN. It should be explained shortly but clearly what they stand for.

I suggest that Bolsa as well as other Brazilian civil, social, health and educational services necessary for understanding this article are presented in Introduction.

According to Abstract, a previously tested structured questionnaire was used in data gathering whereas semi-structured questionnaire is mentioned in the line 89. Please, check this. This has also a connection to the aspect of potential earlier Bolsa evaluations. Has this tested questionnaire been used, and if yes, what where the results then?

Tables 1 and 2 include so many variables and so much information that the reader must really concentrate when studying them. This is even more difficult since the information about perinatal outcomes, the core of the main results, is given only in written form in the beginning of section 3.2. (Bolsa Familia Program and perinatal outcomes).

As the essence of Table 2 is explained in the text, perhaps the table itself could be as an Appendix.

The limitations of this study and suggestions for further research should be added to Discussion. 

All the best for the revisions phase of this important paper!

Author Response

The topic of this article is among the most important ones in human and social perspectives as it deals with the health of pregnant women and newborn babies. The scene is in Brazil where the numbers of infant morbidity and mortality are high. The research reported in the manuscript evaluates the impact of a family program (Bolsa Familia Program BFP) on pregnant women of a named cohort.

The research aim is defined in Introduction as verifying the effects of BFP on perinatal outcomes, including premature birth and low birth weight. It is easy for the reader to understand the basic idea of the article and the main results reported.The research aims as such are clear and from medical point of view relevant literature on perinatal outcomes is used to justify the chosen research approach.

However, other evaluations of Bolsa are not brought forward when justifying this individual research. The reader does not know whether this is a first time, when perinatal outcomes are evaluated as connected to Bolsa. In addition, there are some details that need revision in general, but especially from the point of view of international readership.

Answer: We appreciate the comments, and we performed the requested revision in the introduction. 

The title of the article "Bolsa Familia Program and perinatal outcomes: NISAMI cohort" includes two phenomena that should be defined and explained clearly and thoroughly.

Bolsa Familia Program (BFP) is mentioned shortly in Introduction, but the reader is not provided with all necessary information about it and the services it offers to citizens. Some features of BFP are mentioned only in Discussion section, together with references about other services of which the reader does not know, whether they belong to Bolsa or to other systems. The content of conditionality of Bolsa is not explained at all. Usually, the conditionality in welfare programs like this is supposed to be among the most significant mechanisms that secure the desired impact. Also, Discussion section includes some international references that could be used when presenting the research aim in its social surroundings.

Also, the reader needs to know whether this  program, started in 2003, is still going on. There have been changes in the Brazilian government, raising a question about Bolsa's survival. In this case Bolsa was a real factor in the lives of the pregnant mothers during data gathering and this cannot be changed. What about Discussion section? It should be up to date.

NISAMI is explained in 2. Materials and Methods, 2.1. Study Design. However, it is difficult to understand, what NISAMI actually is. The same can be said about other abbreviations like SIBEC and CadUNICO-SIDUN. It should be explained shortly but clearly what they stand for.

I suggest that Bolsa as well as other Brazilian civil, social, health and educational services necessary for understanding this article are presented in Introduction.

Answer: we insert more information about this  questions in the introdution.

According to Abstract, a previously tested structured questionnaire was used in data gathering whereas semi-structured questionnaire is mentioned in the line 89. Please, check this. This has also a connection to the aspect of potential earlier Bolsa evaluations. Has this tested questionnaire been used, and if yes, what where the results then?

Answer: we insert more information about the questionnaire structure and data collection procedures in the metodology.

Tables 1 and 2 include so many variables and so much information that the reader must really concentrate when studying them. This is even more difficult since the information about perinatal outcomes, the core of the main results, is given only in written form in the beginning of section 3.2. (Bolsa Familia Program and perinatal outcomes).

As the essence of Table 2 is explained in the text, perhaps the table itself could be as an Appendix.

Answer: We insert the table as appendix

The limitations of this study and suggestions for further research should be added to Discussion. 

Answer: We added this information.

Round 2

Reviewer 2 Report

Dear Authors, 

after checking your revised version and new appendix of this paper, I see this it ready to be published.